# Evaluation of Thermally Treated *Calotropis Procera* Fiber for the Removal of Crude Oil on the Water Surface

**DOI:** 10.3390/ma12233894

**Published:** 2019-11-25

**Authors:** Larissa Sobral Hilário, Raoni Batista dos Anjos, Henrique Borges de Moraes Juviniano, Djalma Ribeiro da Silva

**Affiliations:** 1Postgraduate Program in Petroleum Science and Engineering of the Federal University of Rio Grande do Norte, Av. Sen. Salgado Filho, 3000-Lagoa Nova, Natal, RN 59072-970, Brazil; larissasobralhilario@yahoo.com.br (L.S.H.); raonianjos@gmail.com (R.B.d.A.); henriquebm.eng@gmail.com (H.B.d.M.J.); 2Institute of Chemistry of the Federal University of Rio Grande do Norte, Address: Av. Sen. Salgado Filho, 3000-Lagoa Nova, Natal, RN 59072-970, Brazil

**Keywords:** *Calotropis procera*, sorption, crude oil

## Abstract

Biosorbents have been highlighted as an alternative method for the removal of contaminants from spills or leaks of oil and its derivatives, since they are biodegradable, are highly available, low-cost, and have a good sorption capacity. This research investigated the sorption capacity of *Calotropis procera* fiber in natura (CP) and thermally treated (150 °C and 200 °C) for crude oil removal and recovery. The oil sorption tests were carried out in a dry and water (layer) static systems. The assays revealed that CP fiber has excellent hydrophobic-oil properties and good crude oil sorption capacity, about 75 times its own weight (76.32 g/g). The results of the treated fibers, CPT150 and CPT200, showed oil sorption capacities (in 24 h) higher than CP, between 94.31–103.37 g/g and 124.60–180.95 g/g, respectively. The results from sample CPT200 showed that it can be an excellent biosorbent for the removal of crude oil and other derivatives due to its high hydrophobicity, great reuse/resorption capacity, and ability to retain oil within the fiber lumens. Thus, it can be applied in the recovery, cleaning, and removal of petroleum products and its derivatives from spills and leaks in the future.

## 1. Introduction

The increase in the production and transportation of petroleum products in recent years increased the risk of oil spill and chemical leaks, since these organic solvents are toxic and can cause serious damages to the environment [1,2,3,4]. The impacts of oil pollution are unprecedented, since it disturbs marine life, soil, and air, and it is still a threat to human health with repercussions even in tourism and leisure. In addition to compromising the economy [5], because it is a valuable energy resource, it require demanding a fast and effective recovery, attracting interest from researchers [6,7].

During an event of oil leakage, factors such as composition, density, dispersion, emulsification, and evaporation [8] influence the choice of the technique for fast countermeasures to collect and recover the product in order to minimize its pollution potential [9].

Nowadays, the main technologies used for cleaning oil leaks from the water surface are (i) physical diffusion [10], (ii) in situ burning [11], (iii) bioremediation [12], and (iv) mechanical recovery [13]. Among these, it is important to highlight the mechanical recovery with the use of absorbers for cleaning oil spills and insoluble organic solvents in water as an attractive method due to its simplicity and collection integrity not resulting in a secondary pollution [14].

The absorption technology aims to design and develop oil sorbents that have good hydrophobicity, oilphilicity, high absorption capacity, rapid absorption kinetics, biodegradability, and environmental acceptability to absorb the oils from polluted environments [6,14,15,16,17].

Generally, sorbents can be classified into three categories: synthetic polymers, inorganic mineral materials, and natural organic materials [18]. A short time ago, a series of synthetic polymers presented high absorption and reuse capacity such as polyurethane [19], formaldehyde-melamine-bisulfite sodium bisulfite [20], and polyvinylalcohol formaldehyde [21]. Despite this, the non-biodegradability, complex preparation process with many stages, results in a costly and limited production for large-scale industrial manufacturing. Inorganic mineral materials, such as silica [22], zeolites [23], perlite [24], sepiolite [25], and vermiculite [26], present low oil absorption capacity and inadequate buoyancy, and are not efficient in removing oil spills.

On the other hand, organic natural materials, such as fibrous organic sorbents, have a relatively higher sorption capacity, an equal density, or, in some cases, are smaller in comparison to inorganic and synthetic sorbents, besides presenting low-cost, biodegradability, and environmental sustainability [27].

In this sense, with an emphasis on green chemistry, there has been a growing interest and increased research in the development of absorbers materials to clean oil spills in recent years, based on natural fibers also called biosorbents, due to their characteristics and advantages [28,29,30,31].

*Calotropis Procera* (CP) is a shrub from the *Asclepiadaceae* family, which can be found in several regions of the world. It has a natural fiber consisting mainly of cellulose, hemicellulose, lignin, pectin, and wax, as well as hydrophobicity and oleophilic properties. It is biodegradable, light, and has a hollow physical structure [32]. In addition, this fiber has large lumens that contribute to the excellent absorption and retention capacity of oil, making it a promising material for sorption of various oils reported in other studies [33]. Several thermal studies have been conducted to modify characteristics and properties of natural fibers [34,35,36] in order to increase the sorption performance of oils, becauseit is a simple and low-cost method.

Therefore, the objective of this research was to compare the sorption performance of the CP in natura fiber, with the fibers obtained by thermal treatments, for crude oil and testing on other types of oils in water, t so determining that the resulting fiber has a greater capacity and may be used in the future for cleaning, removing, and recovering oils from spills or leaks.

## 2. Materials and Methods 

### 2.1. Material

The *Calotropis procera* (CP) fruits were collected in the coastal region of the Natal municipality, Rio Grande do Norte state, Brazil. The oil used in the sorption experiments was provided by PETROBRAS, Guamaré, Rio Grande do Norte state, Brazil. Moreover, the water of analytical grade was obtained from a reverse osmosis system.

### 2.2. Fiber Thermal Treatment 

Initially, the CP fibers were collected and manually separated from the seeds, being dried at room temperature (25 ± 1 °C) for 24 h. Then, the fibers were thermally treated in a muffle furnace at 150 °C and 200 °C for 1 h, and stored in a plastic container and named according to their treatment temperature, CPT150 and CPT200, respectively.

### 2.3. Fiber Characterization 

The thermal stability of the CP in natura was evaluated by thermogravimetry analyses (TG) and Derivative Thermogravimetry using a thermogravimetric Analyzer from NETZSCH, TG209F1 Libra (Netsch, Selb, Germany). Approximately 7 mg of sample were used in the TG/Derivative Thermogravimetry (DTG) analyses, with a heating rate of 10 °C/min and temperature ranging from 28 to 900 °C, under a dynamic nitrogen atmosphere and a flowrate of 20 mL/min.

The Fourier Transform infrared spectrometry (FT-IR) analysis of the CP, CPT150, and CPT200 fibers were performed using the Frontier instrument (Perkin Elmer, Waltham, MA, USA) at a spectral range between 400–4000 cm^−1^, with a resolution of 4 cm^−1^.

The surface morphology of the CP, CPT150, and CPT200 fibers were characterized with a field emission scanning electron microscope (SEM-FEG), Zeiss Auriga 40 (Zeiss, Oberkochen, Germany), with a power of 15 kV. The fibers were coated with a gold film by sprinklers.

The contact angle measurement (θ) of water and oil in samples CP, CPT150, and CPT200 were performed in a Tensiometer, model K100C (Krüss, Hamburg, Germany).

For the recording of microscopic images regarding the lumen and sorvido oil inside, we used a BIO1B binocular biological optical microscope (Bel Photonics, Monza, Italy).

### 2.4. Selectivity Assay

For the selectivity test of the fiber sorption for oil removal, approximately 1 mL of crude oil was added into 50 mL of distilled water inside a beaker in order to form an oil layer on the water surface. Then, the fibers were placed in contact with the oil layer, aided by a clamp, in order to record the selectivity of sorption. The crude oil used had a specific gravity of 0,861 g/cm^3^ and absolute viscosity of η = 73.6 cP.

### 2.5. Sorption Assay

The CP, CPT150, and CPT200 samples, 10 g each, were immersed in a glass vial containing 5 mL of crude oil, remaining in contact during different time intervals, 5, 20, 40, 60, and 1440 min, at room temperature (25 ± 2 °C) for the static dry test, while for the sorption tests in oil layer in water, the oil/water volume ratio was 1:5. 

After the predetermined times, the fibers were removed with the aid of a clamp, drained on a stainless steel sieve for 5 min, and weighed afterwards. All assays were performed in triplicate under static condition for crude oil, with an absolute viscosity (η) of 73.6 cP. 

From the values of initial and final weight of the fibers, the sorption capacity of the specific medium was calculated using Equation (1) [5]: (1)$$\left(S\right)=\frac{\left(W_{f}-W_{i}\right)}{W_{i}},$$
where:

*S* is the sorption capacity in g of oil/g of the sorbent;

*W*_i_ (g) is the initial weight of the material dry before the oil sorption; ;

*W*_f_ (g) is the final weight of the material after the oil sorption.

The possibility of fiber reuse was evaluated, by compressing the fiber, after oil sorption, being calculated from the weight ratio of the resorption capacity for the initial sorption, and expressed in percentage.

## 3. Results

### 3.1. Characterization of the Calotropis Procera Fibers

#### 3.1.1. Thermogravimetric Analysis—TG 

The TG and DTG analyses were used to evaluate the thermal stability and the CP decomposition with the increase of temperature. After the analysis of the thermogravimetric curves TG/DTG, temperatures for the thermal treatments of CP were determined in order to avoid fiber decomposition. According to Tu et al. (2018), the thermal treatment favors oil sorption capacity. Consequently, it increases the hydrophobicity and oleophilicity of the fiber.

Figure 1 shows the thermal behavior of CP fiber in inert nitrogen atmosphere. One can observe four mass loss events for the TG and DTG curves, the first being between 28–100 °C, with a mass loss of approximately 5.9%, mainly related to water (moisture). In the TG and DTG curves (Figure 1), it was possible to observe that the CP fiber has thermal stability of up to 213 °C.

The second event shows a mass loss of 58.8%, between 213 °C and 354 °C, referring to hemicellulose degradation. According to Yang et al. (2007) [37], the hemicellulose decomposition occurs between 200 and 315 °C. Furthermore, between 354 and 446 °C, one can note a mass loss of 20%, which can be attributed to cellulose degradation. In addition, in the temperature range of 240–400 °C, intense splitting of the cellulose polymeric chains occurs, followed by the onset of lignin decomposition, even with the decomposition of cellulose being the dominant process at this stage [38]. 

The lignin decomposition occurs gradually from temperatures from 100 to 900 °C and happens along with the degradation process of hemicellulose and cellulose. This slow decomposition process can be associated to the complex structure of the aromatic rings that constituted lignin [37]. The degradation peak of the remaining lignin can be observed at high temperatures ranging from 400 to 550 °C [39]. The fourth event observed in the range between 446–516 °C presented a mass loss of 9.76%, referring to lignin.

The study of the thermal stability for *Calotropis procera* (CP) performed by Oun and Rhim (2016) [40] presented only two mass loss events. The first event occurred between 60–120 °C with a mass loss between 4.1–5.6%, mainly due to water loss, and the second and greater mass loss between 200–400 °C, mainly due to degradation of cellulosic materials [41].

#### 3.1.2. FTIR Analysis

Figure 2 shows the FTIR spectra of the CPT150 and CPT200 fibers. The band comprised in the region from 3000 to 3350 cm^−1^ is characteristic of the O–H stretch, corresponding to alcohols, phenols, and carboxylic acids present in the fiber composition [37,40,42]. The peak at 2920 cm^−1^ is attributed to the stretch of bonds C–H, of CH_2,_ and CH_3_ aliphatic, characteristic of vegetable waxes, consisting of N-alkanes, fatty acids, aldehydes, ketones, and esters [42,43]. The peaks between 1630 and 1734 cm^−1^ are characteristic of conjugated and non-conjugated carbonyl (C=O), respectively, probably originating from carboxylic acids and ketones from the hemicellulose and/or lignin groups [27,44]. A symmetrical flexion of CH_2_ and elongation of C–O and C=C characteristic of lignin and cellulose was presented at 1424 cm^−1^. A flexion of C–H is located at 1368 cm^−1^, while vibrations of CH_2_ at 1314 cm^−1^. At 1244 cm^−1^ there is a peak associated with a double bond C=O with stretch, at 1032 cm^−1^ a simple C–O bond from hemicellulose and lignin can be observed, while at 896 cm^−1^ deformation and stretching through C–O–C, C–C–O, and C–C–(H) was detected [45,46,47]. 

When comparing the spectra of the CP fiber with the treated fibers, CPT150 and CPT200, the thermally treated samples showed a decrease in the intensity of the functional groups, including C–H (2920 cm^−1^), C=O (1734, 1368, and 1244 cm^−1^), and C–O (1032 cm^−1^) [48]. According to Tu et al. (2018), such weakening and/or disappearances may be an indication of successful heat treatment [14]. As observed by Draman et al. (2014), attenuations or disappearance of the near peaks corresponding to lignin (1505 and 1597 cm^−1^) and hemicellulose (1737 and 1248 cm^−1^) were noted [49].

The broadband region located between 3350 and 3000 cm^−1^ decreased significantly with temperature increase for the treatments of fibers CP150 (150 °C) and CPT200 (200 °C). In fact, this band is usually associated to OH stretch vibrations and hydrogen bonds of hydroxyl groups, associated with the general binding by intramolecular and intermolecular hydrogen and free hydroxyl in the cellulose macromolecule. However, the absence of a clear structured form makes it difficult to assign this absorption range, as this peak is also representative of the contribution of free water or linked to the substrate [42]. 

#### 3.1.3. Morphological Analysis—SEM-FEG

Figure 3 presents the micrographs of the CP, CPT150, and CPT200 fibers. It is possible to verify (Figure 3) the presence of hollow lumens in the fibers, which allow the fixation of the oil and retention of inter- and intra-fiber structures [14]. Such microstructure can also help in the buoyancy due to the interior empty spaces being filled with air.

The micrograph of CP fiber (Figure 3a) shows a smooth surface and with hydrophobic bristly coating the hollow structure, also reported by Thilagavathi et al. (2018) [50]. However, after being subjected to temperatures of 200 °C, the fiber suffered deformation, presenting a more limpid aspect. Such an event can be attributed to the removal of part of the wax, giving more malleability to the fibers.

According to Kalia et al. (2009), although the natural fibers present mostly smooth surfaces, after the thermal treatments, the fibers present a few deep grooves. These grooves are associated with roughness present in the fiber [51], a characteristic that can favor oil sorption.

Figure 3d–f presents, respectively, the appearance of the fiber in the natural state, with the fiber thermally treated at 150 °C and 200 °C. One can observe an increase in the color intensity of the fiber, a result of the temperature increase used in the heat treatment.

### 3.2. Sorption Capacity

Figure 4a shows the results of the dry sorption tests with a variation of the time from 5 to 1440 min, and as expected, the sorption increases over time. Because the CP fiber has large lumens coated with waxy material, it has a high oil sorption capacity between 48.61 g/g and 74.04 g/g. In order to achieve a material with better oil sorption capacity (oil), it was proposed to remove part of the hydrophobic wax from the fiber surface through heat treatment.

As shown in Figure 4a, the dry sorption capacity in the thermally treated fibers at 150 and 200 °C increased significantly between 15% and 68% from the fiber capacity without treatment. The increase in the temperature from 150 to 200 °C resulted in an increase of the maximum sorption capacity from 94.31 g/g to 124.60 g/g. 

In order to evaluate the water sorption capacity, tests were performed using the CP, CPT150, and CPT200 fibers, presented in Figure 5.

As observed (Figure 5), the increase in the fiber treatment temperature results in the increase of water sorption, as expected, due to the removal of the hydrophobic wax material from the fiber surface. The results of the water sorption tests in samples CP, CPT150, and CPT200 reveals that the water ranged from 0.03 to 0.75 g/g.

In the layer sorption tests (water/oil) the CP, CPT150, and CPT200 fibers presented superior results to the dry tests, with a maximum sorption capacity between 76.32 g/g (CP) and 180.95 g/g (CPT200). The obtained results are shown in Figure 4b. The water sorption values were lower than the layer tests results, being insignificant in relation to the amount of absorbed oil.

Based on that, Figure 4b revealed that the best conditions of the oil sorption capacity was associated to fiber CPT200, contact time of 1440 min, for both dry and layer (static) systems, respectively, 124.60 g/g and 180.95 g/g, the CP sample obtained an approximate adsorption of 75 times its own weight (76.32 g/g).

Studies performed by Tu et al. (2018) confirmed that carbonization improved the oil sorption capacity for the *Calotropis giganteia* crude fiber [14]. Similarly, it was proved that the thermally treated CP considerably favored the oil sorption capacity in relation to CP in natura, obtaining an increment of about 68.28% and 137.09%, respectively, for the dry and layer static system. However, it is observed, in general, that the oil sorption capacity of thermally treated CP fiber, as summarized in Table 1, is comparable and even higher than the most reported sorbents in dry tests. In addition, the CPT200 fiber is produced from its precursor material (*Calotropis procera*) and, consequently, this is a potentially profitable and environmentally friendly material for the removal and recovery of oil in water.

In a generic way, Table 1 shows that the sorption capacity of oils and organic solvents for the sorbent materials that suffered a thermal processes presented better results. It is possible to attest that thermally treated *Calotropis Procera* is a sorbent material that has a high sorption capacity, for the crude oil (viscosity was η = 73.6 cP) used in this research. The characteristics of the oil, such as the viscosity, is a parameter of great importance in the sorption process, because decrease in the viscosity of the oil reduces absorption within the pores and capillary vessels of the materials, and more viscous oils have greater sorption due to adhesion to the surfaces of the materials and inside the pores. Teas et al. (2001) evidenced the role of viscosity in sorption processes, studying how different viscosities of oils promoted varied results, depending on the type of sorbent material that was evaluated [63]. Wei et al. (2003) showed that increased viscosity was able to increase crude oil sorption capacity in different sorbents [9].

Considering the excellent results of this study on the sorption obtained for oil, the sorption of diesel, marine diesel, motor lubricant oil, used lubricating oil, and benzene in the CPT200 conditions compared with CP was also investigated, in the optimum time of 1440 min sorption in both dry and layer systems, as shown in Figure 6.

One can observe in Figure 6a that the dry test for the CPT200 fiber presented a higher sorption capacity (51.75–103.03 g/g) than the CP for all oils and solvents that were tested. Similarly, the same behavior was verified in the layer system, shown in Figure 6b, in which the CPT200 presents sorption from 60.91 to 117.98 g/g.

Karan et al. (2011) [64] highlight in their studies that the oil viscosity is a parameter of great importance in the sorption process, and that the decrease of the oil viscosity reduces the sorption within the pores and capillary vessels. Thus, more viscous oils have higher sorption values due to adhesion on the surfaces of the materials and within the pores. Therefore, Figure 6 shows the predicted results: Lower sorption of diesel, marine diesel, and benzene compared to lubricating oils show that the more viscous the oil, the greater the sorption trend by the greater amount of oil to be sorption by the CP and CPT200 fibers.

Figure 7 shows the possibility of fiber recycling. The resorption capacity of the CPT200 for six cycles was 40.47% oil when compared to the initial sorption. However, until the third cycle the CPT200 obtained a resorption capacity greater than 60%, demonstrating that fiber has reuse potential.

### 3.3. Wettability and Contact Angle

The wettability of the material surface is determined by the value of the contact angle. In order to verify the hydrophobic and hydrophilic properties of CP, CPT150, and CPT200 fibers, the contact angle for water and oil on the surface of the fibers were measured.

As presented in Figure 8 d,e, f, the contact angle for the oil on the surface of fiber CP, CPT150, and CPT200 was 0°, thus, presenting oleophilic characteristics. However, in Figure 8a,b,c, water drops are visible on the surface of CP, CPT150, and CPT200, and the contact angles (θ) of water reached 128°, 119°, and 114°, respectively, demonstrating hydrophobic properties. For CPT150 and CPT200 there was a decrease in contact angles for water, a consequence of heat treatment and the possible decrease in the cerous surface of the fibers. This was also observed in the infrared spectra (Figure 2) by the decrease in the intensity of the region of 2920 cm^−1^, referring to the C–H stretch.

As shown in Figure 8a–c the water drops with blue, green, and orange dye presented spherical shapes on the fiber surface, while the oil droplets (Figure 8d–f) spread immediately to the interior of CP, CPT150, and CPT200, demonstrating its excellent hydrophobic and oleophilic properties.

### 3.4. Selectivity Test

Figure 9 shows the process used in the selectivity test to remove the oil from the water using the CP fiber.

The selectivity test presents the addition of oil into water, forming a film. Then, the CP fiber was placed in contact with the oily surface. One can observe that the CP carried the petroleum, that is, the oil was selective and completely sorption by the fiber. In addition, it was noted that during the test, the CP fiber soaked in oil floats on the surface, not sinking in the water, and thus suggesting an excellent oil sorbent and buoyancy.

### 3.5. Oil Fixation to the Sorbent

In order to investigate the oil sorption in the fibers, an optical microscope was used. With the aid of a microsyringe, oil droplets were scattered on the fiber surface, then microscopic images were recorded, as seen in Figure 10.

Figure 10a presents an empty hollow structure of the lumen by designating to be full of air, allowing the buoyancy of the fiber. In Figure 10b one can observe a fiber with absorbed oil inside by capillary action. In Figure 10c it presents two cross fibers forming a firmly connected oily liquid bridge due to adhesive forces (adsorbent component) and due to strong capillary forces (capillary component).

## 4. Conclusions

In summary, the CP, CPT150, and CPT200 fibers showed high hydrophobicity, oilphilicity, and selectivity for petroleum and derivatives, confirmed by the analysis of the contact angle for water and crude oil, presenting predominantly hydrophobic surfaces, with θ of 128°, 119°, and 114° with water, respectively, and θ of 0° for crude oil on all fibers. Thermogravimetric analysis in CP showed thermal stability of up to 213 °C when the degradation process of the fiber hemicellulose was initiated, followed by cellulose and finally a peak that characterizes lignin decomposition. The FTIR spectra indicated that after the thermal treatments at 150 °C and 200 °C there was a slowdown and disappearance of some peaks, which may be correlated with the partial wax removal. The SEM-FEG micrographs revealed the morphology of the surface CP, CPT150, and CPT200 in which there is a hollow structure with lumens that allows the fixation of the oil. However, after the thermal treatment, some shallow grooves were present. The sorption of CP was about 75 times its weight and CTP150 and CPT200 presented crude oil sorption for the dry and layer test (1440 min) of between 94.31 g/g–103.37 g/g and 124.60 g/g–180.95 g/g, respectively. The CP, CPT150, and CPT200 have an insignificant sorption of water. The performance of sorption tests with other types of oils and solvents in the dry and layer systems with CP and CPT200 resulted in the same behavior profile in which the fiber treated at 200 °C obtained better sorption capacities in both systems when compared to CP, as observed in the significance of viscosity for the sorption process. In this way, CPT200 can be employed as a promising alternative for the removal of crude oil spills and leaks, due to its good oil/water selectivity, great reuse/resorption capacity, high availability, and excellent sorption property of oils and organic solvents.

## Figures and Tables

**Figure 1 materials-12-03894-f001:**
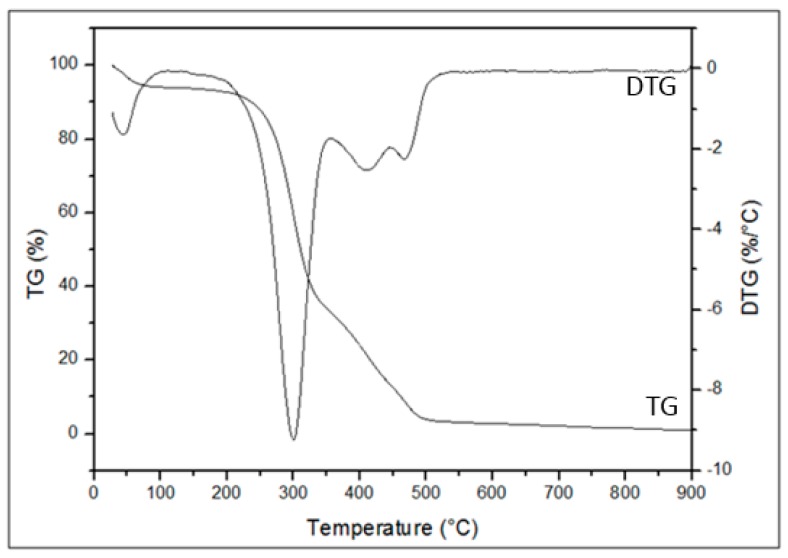
Thermogravimetry (TG) and Derivative Thermogravimetry (DTG) curves of *Calotropis procera* (CP).

**Figure 2 materials-12-03894-f002:**
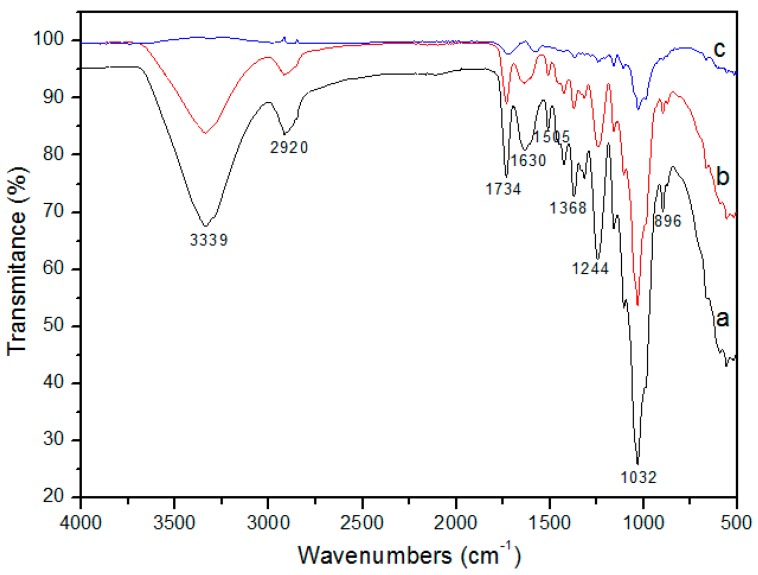
FTIR spectra of (**a**) CP, (**b**) CPT150 and (**c**) CPT200.

**Figure 3 materials-12-03894-f003:**
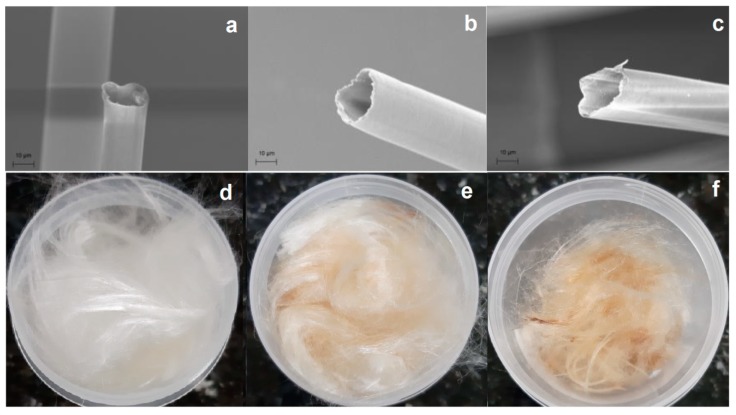
Micrographs obtained by SEM-FEG (**a**) CP, (**b**) CPT150, (**c**) CPT200, (**d**) CP fiber image, (**e**) CPT150 fiber image, and (**f**) CPT200 fiber image.

**Figure 4 materials-12-03894-f004:**
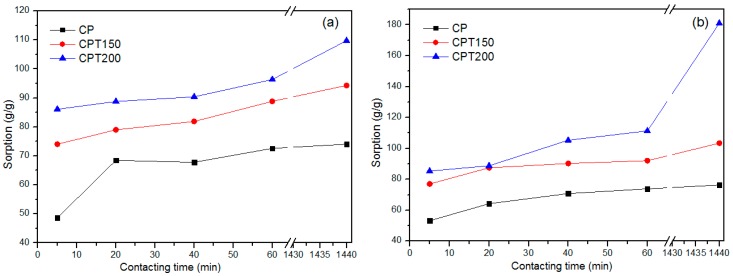
(**a**) Dry and (**b**) crude oil layer sorption test for samples CP, CPT150, and CPT200.

**Figure 5 materials-12-03894-f005:**
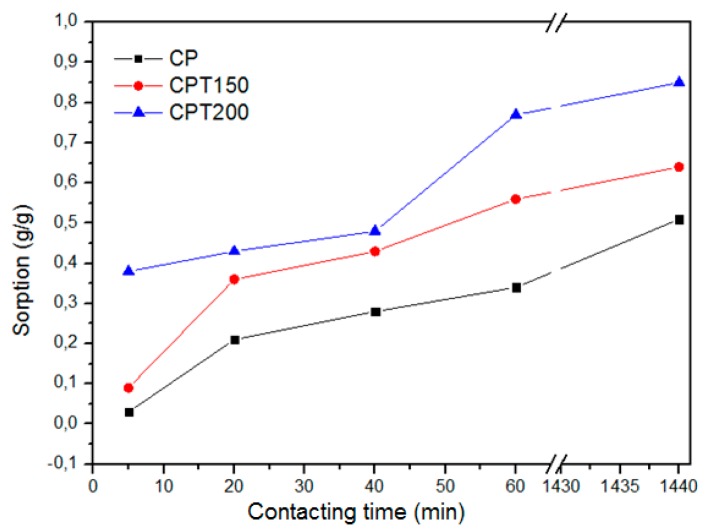
Water sorption test.

**Figure 6 materials-12-03894-f006:**
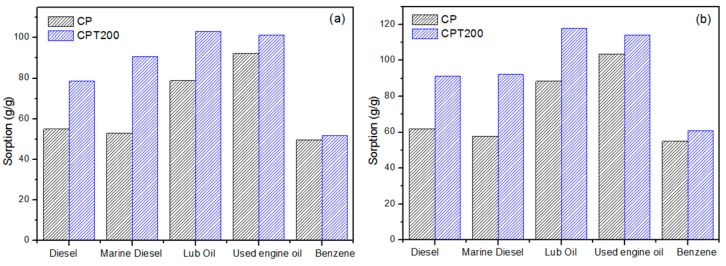
(**a**) Dry and (**b**) layer oil sorption test comparing CP and CPT200 for the time of 1440 min.

**Figure 7 materials-12-03894-f007:**
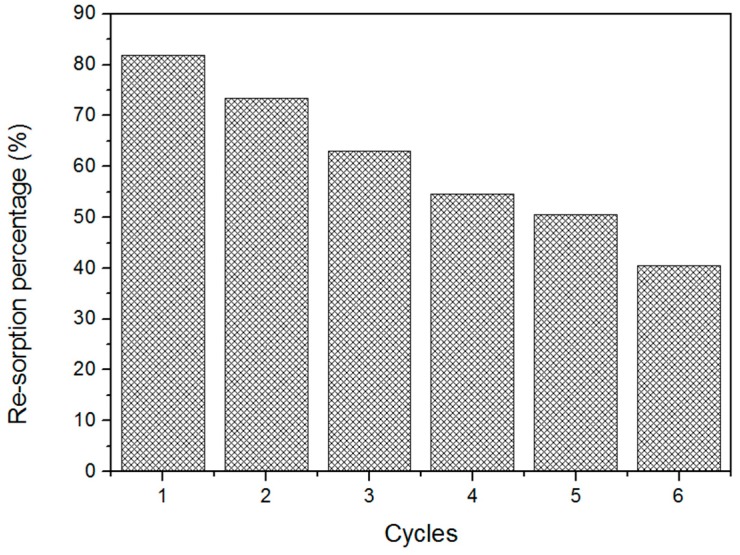
Resorption capacity test for fiber CPT200 in petroleum.

**Figure 8 materials-12-03894-f008:**
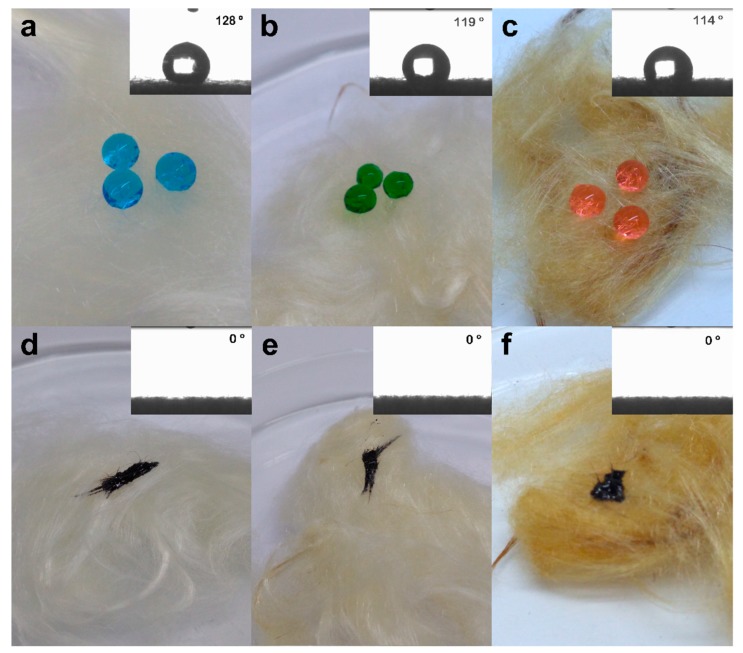
Images of (**a**) water drop with blue dye and contact angle in CP; (**b**) water drop with green dye and contact angle in CPT150; (**c**) water drop with orange dye and contact angle in CPT200; (**d**) crude oil droplets and contact angle in CP; (**e**) crude oil droplets and contact angle in CPT150; (**f**) crude oil droplets and contact angle in CPT200.

**Figure 9 materials-12-03894-f009:**
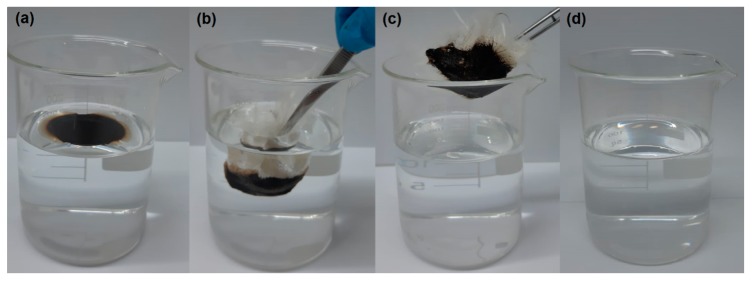
Digital images of oil removal in water (**a–c**) show oil–water separation using CP as sorbent. (**a**) A total of 1 mL of oil was added into 50 mL of distilled water to form an oil layer, (**b**) rapid oil sorption by CP, (**c**) oil removal sorption by CP, and (**d**) clean water after oil sorption.

**Figure 10 materials-12-03894-f010:**
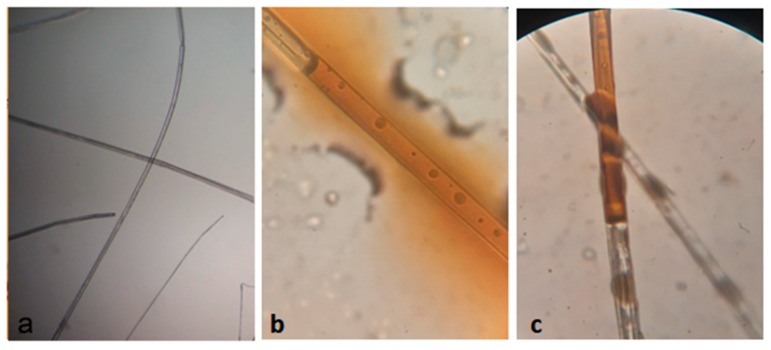
Microscopic images of (**a**) CP fiber—10×; (**b**) oil sorption inside the CP lumen—100×; (**c**) oil trapped between two cross fibers—100×.

**Table 1 materials-12-03894-t001:** Comparison of sorbent materials of oil and organic solvents.

Materials	Treatment	Sorption Capacity—Oil	Author
Sumauma Fiber	Packed	36 g/g—Diesel 43 g/g—Hydraulic oil 45 g/g—Motor oil	[30]
Barley straw	Pyrolyzed	5.9–7.6 g/g—Diesel 8.1–9.2 g/g—Heavy oil	[36]
Silkworm cocoon	Cocoon residues	42–52 g/g—Motor oil 37–60 g/g—Vegetable oil	[52]
Cotton fiber	Loose fiber Fiber pad shape	22.5 g/g—Lubricating oil 18.43 g/g—Lubricating oil	[53]
Peat	Granular	9–12 g/g—Diesel	[54]
Cotton fiber	Carbonized in N_2_ atmosphere	32–77 g/g—Crude oil and solvents	[55]
*Populus* fiber	Acetylation	21.57 g/g—Corn oil	[56]
Clay polymer aerogel	Aerogel	23.6 g/g—Dodecane	[57]
Celulose aerogel	Methyltrimetoxissyan	40–95 g/g—Oil	[58]
*Calotropis gigantea* fiber	In natura	22.6–47.6 g/g—Oil and organic solvents	[59]
Non-polyester fabrics fiber	(Methylhydro-dimetil) siloxane	5.52 g/g—Dodecane 10.03 g/g—Motor oil	[60]
Nanostructured electrospun fibers	Polissulfona/NiFe_2_O_4_	9.20 g/g—Dodecane 15.11 g/g—Motor oil	[61]
*Calotropis gigantea* fiber	Carbonized	80–130 g/g—Oil and organic solvents	[14]
Mix of cotton, Sumauma, *Asclepias Syriaca*, *Calotropis procera*, *Gigantea* Polypropylene	Thermal	40.16 g/g—Heavy oil 23.00 g/g—Diesel	[50]
*Ganoderma applanatum* mushroom	PFOCTS*	1.8–3.1 g/g—Oil	[62]
*Calotropis procera*	In natura	74.04 g/g—Petroleum	This research
*Calotropis procera*	Thermal	124.60 g/g—Petroleum	This research

* Trichloro (1H,1H,2H,2H-perfluorooctyl)silane [62].

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
