# Peer review of "Evaluation of Thermally Treated Calotropis Procera Fiber for the Removal of Crude Oil on the Water Surface"

_materials, 2019, doi:10.3390/ma12233894_

Round 1

Reviewer 1 Report

This manuscript (Ref. No. 632144) deals with Calotropis Procera (CP) natural fibrous material and its thermally-treated derivatives used as oil spill sorbents. The manuscript fits into the scope of the Journal. However, there are some aspects that need to be addressed before considering the manuscript for the next level.

1) In the Title: instead of “..for removal of oil in water”, replace with “..for removal of oil from water surface”

2) As stated in Section 3.1.2 (FTIR), CP is a natural material based on the lignin-hemicellulosic substrate. The hemicelluloses are hydrophilic in nature, and this is obvious from FTIR spectra, where the intensity at 3339 cm-1 (OH) is much greater than the intensity at 2920 (C-H) cm-1, the latter one can be due the wax on fiber.  According to Fig.5, CP is more hydrophobic than CP150 and CP200. Normally, the thermal treatment of natural materials is done to promote the partial pyrolysis of the material in order to increase its hydrophobic properties. It is strange that thermally treated products are more hydrophilic than the pristine material. So, the water sorption test should be verified, because authors are in the range of low uptake capacity (<1 g/g) and the arrangement of fibers in a sample and the mode of contacting with water can make the difference. It would be useful if you can perform the WCA test (water-contact-angle measurements) to see the shape of water droplets on fiber.       

3) On the X-axis of Fig.4 and Fig.5, the authors should specify “Contacting time (min)”; because on testing oil spill sorbents the “drainage time” is also involved as the operating parameter.  

4) How about the recovery test? The spent sorbent (loaded with) oil can be subjected to simple-squeezing or centrifugation in order to recover the retained oil and partially to regenerate the sorbent.   

5) How about the mechanism of oil sorption and retention into the fibrous matrix? Can you record some optical-microscopic images to see the capillary bridges/meniscus between fibers (capillary phenomena).  See for example https://doi.org/10.1016/j.polymertesting.2017.02.024

6) Table 1 should be extended by adding and other types of oil spill sorbents reported in the literature  (such as peat, clay-based aerogels, treated microfibers, and nanostructured electrospun-fibers), see for example:

https://doi.org/10.1016/j.colsurfa.2011.05.036

https://doi.org/10.1016/j.seppur.2014.06.059

https://doi.org/10.1016/j.polymertesting.2017.02.024

https://doi.org/10.1016/j.jtice.2016.11.005

Reviewer 2 Report

This manuscript (Ref. No. 632144) deals with Calotropis Procera (CP) natural fibrous material and its thermally-treated derivatives used as oil spill sorbents. The manuscript fits into the scope of the Journal. However, there are some aspects that need to be addressed before considering the manuscript for the next level.

1) In the Title: instead of “..for removal of oil in water”, replace with “..for removal of oil from water surface”

2) As stated  in Section 3.1.2 (FTIR), CP is a natural material based on lignin-hemicellulosic substrate. The hemicelluloses are hydrophilic in nature, and this is obvious from FTIR spectra, where the intensity at 3339 cm-1 (OH) is much greater than the intensity at 2920 (C-H) cm-1, the latter one can be due the wax on fiber.  According to Fig.5, CP is more hydrophobic than CP150 and CP200. Normally, the thermal treatment of natural materials is done to promote the partial pyrolysis of the material in order to increase its hydrophobic properties. It is strange that thermally treated products are more hydrophilic than the pristine material. So, the water sorption test should be verified, because authors are in the range of low uptake capacity (<1 g/g) and the arrangement of fibers in sample and the mode of contacting with water can make the difference. It would be useful if you can perform the WCA test (water-contact-angle measurements) to see the shape of water droplets on fiber.       

3) On X-axis of Fig.4 and Fig.5, the authors should specify “Contacting time (min)”; because on testing oil spill sorbents the “drainage time” is also involved as operating parameter. 

4) How about the recovery test? The spent sorbent (loaded with) oil can be subjected to simple-squeezing or centrifugation in order to recovery the retained oil and partially to regenerate the sorbent.  

5) How about the mechanism of oil sorption and retention into the fibrous matrix? Can you record some optical-microscopic images to see the capillary bridges / meniscus between fibers (capillary phenomena).  See for example: https://doi.org/10.1016/j.polymertesting.2017.02.024

Reviewer 3 Report

Authors tested efficiency of Calotropis procera fiber as biosorbents for oil removal from oil. This fiber is not novel for this use. Please see: Nascimento et al., 2016 https://link.springer.com/chapter/10.1007/978-94-017-7515-1_9  “Removal of Crude Oil Using a New Natural Fibre—Calotropis procera” Authors should justify their study and mention what is new in this manuscript. Authors use thermally treated fiber but did not give any justification why it can be useful to thermally treat the fiber.

 It would be interesting if authors test the oils with different stages of weathering. During an oil spill, oils go in to weathering as soon as they are exposed to the environment.

Line 46: space after simplicity.

Line 75: One of your goal is to test thermally treated fiber. However, you should mention the advantages (if any) of thermally treated fiber in the introduction. Calotropis procera fiber’s absorbing capacity increases when thermally treated? You can hypothesize at least.

Line 92: spell out TG and DTG for their first use.

Line 107: There are literally hundreds different types of oil. You should give the details of the oil used. Specific information like its density, whether weathered or not are crucial.

Line 107: You mean “Beaker” not becker. Right?

Line 113: When you say “petroleum”, it has wide range of products. What exactly you are using?

Line 120: I saw the viscosity information. Please add this information and other details in line 107.

Figure 1: Please label TG and DTG curve.

Line 130: What is the purpose of TG? How is it related to its oil absorbance capacity? For example, FTIR analysis showing fiber’s functional groups which are important for oil binding and related to oil absorbance capacity.

Line 200-203: This is your justification for using a thermally treated fiber. This should be mentioned in the introduction.

Line 239: 124.60 g/g and 180.95 g/g. Right now it reads as 12460 and 18095.

Table 1: When you are comparing with Reference 47, noticed that in your study thermally treated fiber can absorb more (124.60 g/g) compared to 40.16 g/g. Why you think in your study fiber can absorb more? My guess is it’s due to different oil type. You didn’t mention what type of oil you used. So, its hard to compare with others. Also, fix the number for reference 54 and your study. They should be 1.8 not 1,8 and same for the other numbers.

Line 259: You should mention in your goal that you are testing different type of oil to investigate the efficiency of absorbance of the fiber.

This fiber was used before for oil removal from water. However, authors are also using thermally treated fiber and showed increased efficiency. In my opinion, this manuscript should bring this fact to the front.

Round 2

Reviewer 1 Report

After revision, the manuscript has been improved. 

It can be now accepted for publication, but after addressing several minor corrections related to some typo-errors:  

P11L308 (Page 11, Line 308): "..oil sorcation in the fibers" to be replaced with "..oil sorption in the fibers"

P11L315 : "..with sorvious oil inside" to be replaced with "..with absorbed oil inside"

in Table 1: use "Peat" instead of "Turf"

in Table 1: use "Nanostructured electrospun fibers" instead of "Nanostructured electromagnetic fibers"

Author Response

Response to Reviewer 1 Comments

Point 1: P11L308 (Page 11, Line 308): "..oil sorcation in the fibers" to be replaced with "..oil sorption in the fibers

Response 1: ok, done!

Point 2:P11L315 : "..with sorvious oil inside" to be replaced with "..with absorbed oil inside"

Response 2: ok, done!

Point 3: in Table 1: use "Peat" instead of "Turf"

 Response 3: ok, done!

Point 4: in Table 1: use "Nanostructured electrospun fibers" instead of "Nanostructured electromagnetic fibers"

Response 4: ok, done!

Reviewer 2 Report

The article "Evaluation of thermally treated Calotropis procera fiber for oil removal in water" can be published only after the following problems have been solved:

abstract: line 19 should be reformulated;

In Fig 1 should be presented not only CP but also CPT150 and CPT200;

Author Response

Response to Reviewer 2 Comments

Point 1: abstract: line 19 should be reformulated;

Response 1: ok, done!

Point 2: In Fig 1 should be presented not only CP but also CPT150 and CPT200;

Response 2: Because, the TG and DTG analyses were used to evaluate the thermal stability and the CP decomposition with the increase of temperature. After the analysis of the thermogravimetric curves TG/DTG, it was determined the temperatures for the thermal treatments of CP in order to avoid fiber decomposition.

Reviewer 3 Report

Authors have satisfactorily responded to the comments. This manuscript is ready for publication.

Author Response

Response to Reviewer 3 Comments

Point 1: Authors have satisfactorily responded to the comments. This manuscript is ready for publication.

Thank you for your contributions to improving this manuscript!